

# Roles of oceanic ventilation and terrestrial outflow in the atmospheric non-methane hydrocarbons over the Chinese marginal seas

Jian Wang[1], Lei Xue[2], Qianyao Ma[1], Feng Xu[1], Gaobin Xu[1], Shibo Yan[1], Jiawei Zhang[3], Jianlong Li[1], Honghai Zhang[1, *], Guiling Zhang[1], Zhaohui Chen[4]

[1]Frontiers Science Center for Deep Ocean Multispheres and Earth System, Key Laboratory of Marine Chemistry Theory and Technology, Ministry of Education, and College of Chemistry and Chemical Engineering, Ocean University of China, Qingdao 266100, China

[2]Department of Chemistry, College of Environmental Science and Forestry, State University of New York, Syracuse, NY, 13210, United States

[3] Eco-Environmental Monitoring and Research Center, Pearl River Valley and South China Sea Ecology and Environment Administration, Ministry of Ecology and Environment, Guangzhou 510610, China

[4]Key Laboratory of Physical Oceanography, Ministry of Education, Ocean University of China, Qingdao 266100, China

* To whom correspondence should be addressed: honghaizhang@ouc.edu.cn



**Abstract**
Non-methane hydrocarbons (NMHCs) in the marine atmosphere have been
extensively studied due to their important roles in regulating the atmospheric chemistry
and climate. However, very little is known about the distribution and sources of
NMHCs in the lower atmosphere over the marginal seas of China. Herein, we
characterized the atmospheric NMHCs (C2-C5) in both the coastal cities and marginal
seas of China in spring 2021, with a focus on identifying the sources of NMHCs in the
coastal atmosphere. The NMHCs in urban atmospheres, especially for alkanes, were
significantly higher compared to that in marine atmosphere, suggesting that terrestrial
NMHCs may serve as an important reservoir/source of the marine atmosphere. A
significant correlation was observed between the alkane concentrations and the
distances from sampling sites to the nearest land or retention of air mass over land,
indicating that alkanes in the marine atmosphere are largely influenced by terrestrial
inputs through air-mass transport. For alkenes, a greater impact from oceanic emissions
was determined due to the lower terrestrial concentrations, short atmospheric lifetime,
and substantial sea-to-air fluxes of alkenes compared to alkanes ($489 \pm 454$ vs $129 \pm$
$106$ nmol m$^{-2}$ d$^{-1}$). As suggested by the positive matrix factorization, terrestrial inputs
contributed to 89 % of alkanes and 69.6 % of alkenes in Chinese marginal seas,
subsequently contributing to 84 % of the ozone formation potential associated with C2-
C5 NMHCs. These findings underscore the significance of terrestrial outflow in
controlling the distribution and composition of atmospheric NMHCs in the marginal
seas of China.
**Keywords:** non-methane hydrocarbons, oceanic ventilation, terrestrial outflow, source
apportionment



## 1 Introduction

Non-methane hydrocarbons (NMHCs), a significant subset of volatile organic compounds (VOCs), are acknowledged as key precursors to tropospheric ozone formation (Houweling et al., 1998; Solomon et al., 2005) and second organic aerosol (SOA) generation (Hallquist et al., 2009; Wu and Xie, 2018), playing a pivotal role in atmospheric chemistry. The presence and activity of NMHCs in the troposphere have far-reaching implications, not only influencing the dynamics of ozone and SOA formation but also significantly impacting air quality. These compounds are intricately linked to heightened human health risks as well as possessing indirect yet profound effects on the broader climate system through their interactions with various atmospheric processes (Yuan et al., 2018).

The emission of NMHCs into the atmosphere stems from an array of natural and anthropogenic processes. Oceanic sources of NMHCs predominantly entail the biogenic production of phytoplankton and photochemical degradation of dissolved organic matter (DOM) (Bonsang et al., 1992; Li et al., 2019; Riemer et al., 2000; Sahu et al., 2010). However, they are minimal when compared to terrestrial inputs. Previous estimates driven from a global VOCs emissions model formulated by Guenther et al. (1995) assign terrestrial sources a cumulative emission of 1141 Tg C year$^{-1}$, with oceanic emissions at merely 5 Tg C year$^{-1}$. A substantial amount of NMHCs originating from terrestrial sources (e.g., vehicular emissions, biomass combustion, industrial activities, and continental vegetation emissions) can be transported into the offshore atmosphere via air mass conveyance (Wang et al., 2005; Kato et al., 2007; Song et al., 2020). Subsequently, these supplementary terrestrial NMHCs will play a pivotal role in shaping the chemical composition of the offshore atmosphere and influencing local



environmental dynamics. Hence, to further understand the characteristics, variation,
and origins of NMHCs in the offshore atmosphere, it is imperative to scrutinize oceanic
emissions and meanwhile, it is necessary to figure out the effect of terrestrial outflow
on nearshore NMHCs.

The Yellow Sea and the East China Sea are important parts of Chinese marginal seas,

situated along the eastern coast of China where it is densely populated and has intensive
industries. The rapid pace of Chinese development has seen a notable escalation in
anthropogenic NMHCs emissions over recent decades (He et al., 2019). Presently,
excessive NMHCs emissions and severe ozone pollution have emerged as urgent
environmental challenges in China, particularly in highly urbanized and industrialized
areas along the eastern coast (Liu et al., 2016; Zhang et al., 2018). The seasonal cycle
of the Asian monsoon and diurnal fluctuations of sea-land breezes can facilitate the
transport of terrestrial pollution to the marine atmosphere (Ding et al., 2004; Wang et
al., 2003; Talbot et al., 2003; Russo et al., 2003). Additionally, eutrophication in coastal
regions fosters the proliferation of phytoplankton, potentially augmenting the natural
emissions of NMHCs. Consequently, conducting atmospheric investigations in the
coastal region of eastern China is effective in revealing the potential effects of land-sea
interactions on offshore atmospheric NMHCs.

In the spring of 2021, atmospheric samples were systematically collected from both

coastal cities and marginal seas of China, providing representative insights into the
characteristics of NMHCs (C2-C5) and facilitating discussion on the interplay between
ocean emission and terrestrial outflow concerning atmospheric NMHCs. Ultimately,
the contributions of diverse sources to NMHCs were quantified using the positive
matrix factorization (PMF) model, with assistance from indications provided by other
typical gases, mainly dimethyl sulfur (DMS), volatile halogenated compounds (VHCs),



and monocyclic aromatics.

## 2 Methods

### 2.1 Samples collection

The urban samples were collected from eight coastal cities in China from March 27 to April 1, 2021 (Fig. 1). Air samples were collected using fused silica-lined canisters (2.5 L), which were cleaned three times via a Canister Cleaning System (2101DS, Nutech) and were pumped into a negative pressure state before sampling. The sampling sites were selected at the top of high buildings to minimize contamination of particular point sources. Air samples were collected at 09:00 and 21:00 local time (UTC+8) aimed to represent the urban atmospheric conditions during the daytime and night, respectively. Note that night samples in Xiamen and Qinzhou were missing.

Oceanic air samples were collected aboard RV "*Dong Fang Hong 3*" during the voyage in the Yellow Sea and the East China Sea from April 17 to May 2, 2021. Nineteen oceanic air samples were collected on the top deck facing the wind when the ship was about to arrive at the station and started to slow down. Seawater samples were collected via prewashed Niskin bottles (12 L) incorporated into the Conductivity-Temperature-Depth Sensor Rosette (Seabird 911). Sampling details for urban and marine samples are shown in Table S4 and Table S5, respectively.

### 2.2 Analysis of air samples

All air samples were processed immediately after being brought back to the laboratory, using an Atmospheric Pre-concentrator System (8900DS, Nutech) coupled with a GC-MSD system (GC-7890A, MSD-5975, Agilent). The pretreatments of air



samples were as follows. First, the Atmospheric Pre-concentrator System was baked for
10 min to clean the interior instrument. Then trap 1 was cooled to -170 °C using liquid
$N_2$ and a 300 mL air sample was pumped from the canister into trap 1 for the initial
concentration of the target compounds, while $N_2$ and $O_2$ escaped due to their lower
boiling points. After trap 2 was cooled to -50 °C, trap 1 was heated to 30 °C to transfer
the target compounds from trap 1 to trap 2. Moisture and $CO_2$ were removed in the
second concentration. Then, trap 2 was warmed up to transfer the target compounds
into the last trap for cryofocusing ($-175$ °C). Finally, the last trap was instantaneously
heated to 200 °C via gas bath heating, and the target compounds were delivered into
the GC-MSD system by ultra-pure He.

For the parameter settings of GC-MSD, the temperature of the inlet, quadrupole, and

ionization source was 150 °C, 150 °C, and 230 °C, respectively. The inlet was set to
split mode with a ratio of 10:1. The flow rate of carrier gas (He) was set to 1.5 mL min$^{-1}$
in the instant flow mode. Specific columns were selected to separate the NMHCs (Rt-
Alumina BOND/KCl, Restek), monocyclic aromatics (DB-624, Agilent), DMS
(CP7529, Agilent), and VHCs (DB-624, Agilent). Gas standards of NMHCs in $N_2$
(Linde Gases, Germany) were diluted to 0.1-1 ppb for identification and calibration.
Details of temperature programming and detector parameters can be seen in Zou et al.
(2021) and Li et al. (2019). The precision and detection limits for the trace gases in the
present study were 1-7 % and 0.03-20.0 ppt, respectively (Table S1). Note that DMS
and VHCs data in marine atmospheric samples were graciously provided by colleagues
in the same laboratory. These data were only used as supporting information in the
interpretation (e.g., correlation analysis) of our core dataset in this paper.



**2.3 Analysis of seawater samples**

C2-C5 NMHCs in seawater were measured immediately on board using a purge and trap system coupled with the gas chromatography equipped with a flame ionization detector (GC-FID, 7890B, Agilent). The purge and trap system was improved based on a previously self-designed device described by Li et al. (2019). Briefly, seawater was collected using a customized glass sampler (500 mL) and was connected to the inlet of the system. Then seawater was transformed into the extraction cell under the pressure of pure $N_2$ and was purged with pure $N_2$ bubble flow (250 mL min$^{-1}$). The moisture of the carrier gas condensed in a thin glass tube that was placed in a cold chamber (4-6 °C) and the carbon dioxide was absorbed by the glass tube filled with Ascarite II (Merck). The targets were concentrated in a passivated stainless-steel tube immersed in liquid nitrogen for 26 mins. Then, the steel tube was heated by boiling water and immediately, the six-way valve was turned for the inlet situation. The concentrated target compounds were transferred into the Rt-Alumina BOND/KCl capillary column for separation and were determined by the FID. The parameters of the inlet, oven, and detector are shown in Table S2. The gas standard (Linde Gases, Germany) was diluted with ultra-pure $N_2$ to 10 ppb for identification and quantification. The instrumental blank was made to guarantee data reliability. The precision and detection limits were 3-6 % and 0.5-1.0 pmol L$^{-1}$ (Table S3).

**2.4 Calculation of sea-to-air flux**

The sea-to-air flux of each NMHCs ($F$, nmol m$^{-2}$ d$^{-1}$) was calculated using Eq. (1):

$$F = k \times (C_w - C_a \times H) \tag{1}$$

where $k$ (m s$^{-1}$) is the gas transfer velocity described by Eq. (2); $H$ is Henry's law constant; $C_w$ (pmol L$^{-1}$) and $C_a$ (ppb) are concentrations of each NMHCs in the



surface seawater (5 m depth) and atmosphere, respectively.
$k = 0.31 \times u^2 \times (\frac{Sc}{660})^{-0.5}$                                  (2)
where $u$ (m s$^{-1}$) is the wind velocity at 10 m. $Sc$ is the Schmidt number and is defined
as Sc = $v/D$. $v$ was the kinematic viscosity of seawater calculated by Eq. (3)
(Wanninkhof, 1992). $D$ is the gas diffusion coefficient related to temperature described
by Eq. (4) (Wilke and Chang, 1955).
$v = 1.052 + 1.300 \times 10^{-3} \times t + 5.000 \times 10^{-6} \times t^2 + 5.000 \times 10^{-7} \times t^3$     (3)
$D = \frac{7.4 \times 10^{-8} (q \times M_b)^{0.5} \times T}{n_b \times V_a{}^{0.6}}$                          (4)
where $t$ (℃) is the degree Celsius of seawater, $q$ is the association factor of water,
$M_b$ (g mol$^{-1}$) is the molar weight of water, $T$ (K) is the degree Kelvin of seawater, $n_b$
is the dynamic viscosity of seawater and $V_a$ is the molar volume at the boiling point.
**2.5 Normalized concentrations and lifetime-weighted concentrations of NMHCs.**
To effectively compare the NMHCs variation with respect to the distance from the
sampling sites to the land (like Fig. 2d, f), we calculated the normalized concentration
for each NMHCs ($C_{Nor-i}$) using Eq. (5).
$C_{Nor-i} = \frac{C_i}{C_{max-i}}$                                          (5)
where $C_i$ is the concentration of gas $i$ and $C_{max-i}$ is the maximum of gas $i$.
A novel approach was employed to analyzed the correlation between the
concentrations of various NMHCs and their sea-to-air fluxes. Concentrations were
weighted according to the respective atmospheric •OH lifetime of each NMHCs. This
was achieved by dividing the concentration of each NMHCs by its corresponding
atmospheric •OH lifetime, yield a "lifetime-weighted concentration" for each NMHCs
($C_{life-i}$) (Eq. 6). This method provides a more nuanced understanding of the impact of





oceanic emission on NMHCs, taking into account not only their abundance but also
their residence in the atmosphere.
$C_{life-i} = \frac{C_i}{\tau_i}$ (6)
where $C_i$ is the atmospheric concentration of gas $i$, $\tau_i$ is the •OH lifetime of gas $i$.
Approximate atmospheric lifetime of each NMHCs was calculated assuming an
average [•OH] of $6\times10^5$ molecules cm$^{-3}$ within 24 h at 288 K (Jobson et al., 1999), with
specific data listed in Table 1.
**2.6 Calculation of retention of air mass over land**
To identify whether an air mass was mainly from terrestrial or oceanic regions, the
retention ratio of the air mass over land ($R_L$) was calculated by Eq. (7).
$R_L = \frac{\sum_{n=1}^{N_{land}} e^{-\frac{t_n}{48}}}{\sum_{n=1}^{N_{total}} e^{-\frac{t_n}{48}}}$ (7)
Where $N_{total}$ is the total number of trajectory endpoints (downloaded from NOAA
Air Resources Laboratory HYSPLIT trajectory model https://www.arl.noaa.gov/).
$N_{land}$ is the total number of trajectory endpoints located over land, while $t_n$ is the
backward tracking time with the unit of hour and $e^{-\frac{t_n}{48}}$ is the weighting factor related
to tracking time as the diffusion of air mass takes place along the transport path than in
the nearby regions. As a result, the larger $R_L$ value indicates that the air mass is more
influenced by terrestrial transport and its source is more likely to be on land. Similar
methods have been used to calculate the average residence time of sampled air masses
in the Arctic (Willis et al., 2017) and identify the percentage of time spent by trajectories
over different surface types in the Antarctic (Decesari et al., 2020). $R_L$ values were
calculated by three different time-scale trajectories (48h, 72h, and 96h). The mean $R_L$
(n = 3) was finally applied to analyze the terrestrial influence on oceanic NMHCs,



mitigating the uncertainty caused by the trajectory with different time-scales.

**2.7 Application of the PMF model**

PMF model introduced in detail in the study of (Paatero and Tapper, 1994) was
applied to analyze the data of atmospheric NMHCs in the Yellow Sea and South China
Sea. Based on a matrix consisting of the concentrations of diverse chemical species, the
objective of PMF is to determine the number of NMHCs source factors, the chemical
composition profile of each factor, and the contribution of each factor to species. In the
application of the PMF model, the significance of missing data in the matrix was
decreased by using the species median. The uncertainty for normal data was estimated
as 20 % of the NMHCs concentrations because the analytical uncertainty was not
available (Buzcu and Fraser, 2006). In this analysis, the model ran 20 times and we
selected the result with the minimum "Q value". Besides, approximately 94 % of the
scaled residuals given by PMF ranged from -3 to 3 (Fig. S1), suggesting a reasonable
fit of the model result.

**3 Results and discussion**

**3.1 Atmospheric concentrations of NMHCs in coastal cities and coastal seas of China**

To clarify, NMHCs determined in this study were separated into two groups for
further discussion based on their distinctly different atmospheric reactivity and lifetimes:
alkanes (long lifetime, 8.2-78 d) and alkenes (short lifetime, 0.19-2.3 d). In urban
atmosphere (n = 14), the mean (range) concentration of ethane, propane, i-butane, and
n-butane was 2.26 (0.277-5.72), 2.95 (0.149-20.1), 2.57 (BD-27.6), and 3.29 (0.018-


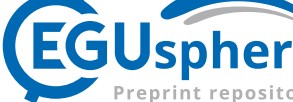

30.2) ppb, respectively (Table 1). Alkanes combined accounted for ~76 %-99 % of total
NMHCs measured in this study, which agrees with previous studies reporting alkanes
as the dominant NMHCs in the urban atmosphere of China e.g., 43.7 % (Song et al.,
2007), and > 50 % (Li et al., 2015). For alkene species in the urban atmosphere (n =
14), the mean (range) of ethylene, propylene, and isoprene was 0.180 (0.035-0.390),
0.036 (BD-0.129), and 0.046 (0.006-0.250) ppb, respectively.
Similarly, alkanes were also dominant components in the marine atmosphere,
accounting for ~86 %-95 % of NMHCs. In the marine atmosphere (n = 19), the mean
(range) concentration of ethane, propane, i-butane, n-butane, ethylene, propylene, and
isoprene was 1.24 (0.686-1.72), 0.822 (0.226-1.79), 0.283 (BD-1.17), 0.256 (0.025-
0.694), 0.151 (0.028-0.295), 0.033 (0.022-0.060), and 0.008 (BD-0.043) ppb,
respectively. These values were comparable to those reported in the Bengal Bay (Sahu
et al., 2011) and the Northwest Pacific Ocean (Li et al., 2019) (Table S6). Alkanes in
the urban atmosphere were on average more than four times higher than those in the
marine atmosphere, while no significant difference was observed for concentrations of
alkenes between urban and marine air ($t = 2.224$, $p = 0.156$) (Fig. 2a, b). This suggests
that the terrestrial alkanes may potentially serve as a reservoir/source of the alkanes in
the marine atmosphere through the transport of terrestrial air mass.

## 3.2 Atmospheric NMHCs variability vs. estimated lifetime

The standard deviation of the natural logarithm of the NMHCs mixing ratios ($S_{lnx}$)
was established to correlate to their •OH lifetime ($\tau$) in the atmosphere following an
exponential function of $S_{lnx} = A\tau^{-b}$ (Jobson et al., 1998), where $A$ and $b$ are fitting
parameters. A $b$ value approaching zero suggests that the NMHCs variability is
primarily controlled by local emission fluctuations while a $b$ value of 1 indicates the





minimal impact of local emissions, with the variability predominantly controlled by the
extent of photochemical reactions.

Employing the analytical framework in Jobson et al. (1998), we analyzed our

atmospheric NMHCs data from urban areas and the Chinese marginal seas. The derived
$b$ value for urban areas was 0.05 (Fig. 3a), suggesting that atmospheric NMHCs in
coastal cities were mainly controlled by local emissions. In the marine atmosphere, the
$b$ value was 0.26 (Fig. 3b) which was comparable to values reported for Gosan (0.30)
(Wong et al., 2007) and continental outflow from southern China (0.31) (Wang et al.,
2005), but it was significantly lower than the values for Ogasawara (0.43) (Kato et al.,
2004), the Northwest Indian Ocean (0.40) (Warneke and De Gouw, 2001), and the South
China Sea (0.42) (Wang et al., 2005). The $b$ value of 0.26 in the atmosphere over the
Chinese marginal suggests that the NMHCs composition in nearshore atmosphere is
influenced both by local oceanic emissions and the remote sources from the continent.
As sites closer to the source position tend to have lower $b$ values, the Yellow Sea and
the East China Sea experience a more pronounced influence from terrestrial pollution
sources compared to Ogasawara, the South China Sea, and the Northwest Indian Ocean.
**3.3 Terrestrial influence on marine atmospheric NMHCs variation**

Given the discernible impact of terrestrial input on the spatial distributions and

variabilities of marine atmospheric NMHCs, we further elucidated the role of terrestrial
outflow in shaping marine atmospheric NMHCs levels. This examination focused on
three key factors: distance from the sampling site to the land, retention of air mass over
land, and transport time of air mass.





**Distance from the sampling site to the land**


The distances from the oceanic sampling sites to the nearest land spanned from 13.9
to 331 km, with an average of 123 km (Table S9). Significant correlations were
observed between the distances and concentrations of ethane ($r$ = -0.553, $n$ = 19, $p$ =
0.014), propane ($r$ = -0.605, $n$ = 19, $p$ = 0.006), i-butane ($r$ = -0.513, $n$ = 19, $p$ = 0.025),
and n-butane ($r$ = -0.573, $n$ = 19, $p$ = 0.010). When plotted against the distances, the
concentrations of alkanes combined decreased with the increasing distance (Fig. 2c),
and different species exhibited distinctly specific decreasing rates (Fig. 2d). Since the
concentrations between different NMHCs species varied considerably, the normalized
concentrations were employed to fit an attenuation equation ($y$ = $Ae^{-tx}+y_0$) for each
species. As evident in Fig. 2d, the attenuation coefficients for ethane, propane, i-butane,
and n-butane were 0.003, 0.030, 0.031, and 0.022, respectively. These coefficients were
correlated with their atmospheric reactivities. Species with lower reactivity and longer
lifetimes, such as ethane (with a lifetime of 78 d), have the lowest attenuation
coefficient. This implies that long-lifetime species could be affected by the terrestrial
input even at a more remote marine site. Terrestrial influences on propane, i-butane,
and n-butane were discernible only in areas much closer to land, as their concentrations
stabilized at low values beyond a distance of around 100 km (Fig. 2d).

**Retention of air mass over land**


A larger retention of air mass over land ($R_L$) has previously been suggested to serve
as an indicator of a greater terrestrial influence (Zhou et al., 2021). To mitigate the
uncertainty derived from varying time-scale trajectories, we calculated the $R_{L-mean}$
based on 48, 72, and 96-hour backward trajectories. $R_{L-mean}$ ranged from 0.10 to 0.96
(Table S9). When plotted against $R_{L-mean}$, a linear relationship was observed between



the concentrations of NMHCs combined and $R_{L-mean}$, with a slope of 2.51 (Fig. 4a).
A statistically significant correlation ($r = 0.599$, $n = 19$, $p = 0.007$) was observed when
only plotting alkanes with $R_{L-mean}$. However, the correlation between alkenes and
$R_{L-mean}$ was statistically insignificant ($r = 0.248$, $n = 19$, $p = 0.306$).
**Transport time of air mass**
The transport time of air mass was estimated as the interval from the last point of the
trajectory contacting the continent to the moment when the air mass reached the
sampling location, as detailed by Kato et al. (2001). These times ranged from 4 to 81 h,
with an average of 30 h (Table S9). A shorter air mass transport time signifies a stronger
terrestrial influence, as NMHCs within the air mass undergo further oxidation and
dispersion over time. Total NMHCs concentrations exhibited a significant decrease
with the increase of air mass transport time, characterized by a slope of -0.04 (Fig. 4d).
Alkanes displayed a steeper decline, indicated by a slope of -0.0079 (Fig. 4e) compared
to alkenes (-0.0038, Fig. 4f). However, similar to the analysis of $R_L$, the correlation
between the air mass transport time and alkenes was statistically insignificant ($r = 0.248$,
$n = 19$, $p = 0.306$).
Overall, the analysis above suggests that the terrestrial input plays an important role
in driving the variability observed for the atmospheric NMHCs over the marginal seas
of China. In particular, a stronger terrestrial impact was determined for the alkanes
based on the larger slopes from linear regression analysis and the significant
correlations with terrestrial indicators. In contrast, no discernible trend was found for
alkenes when plotting their concentrations against the distance from sampling sites to
the coastline (Fig. 2e, f). There was no significant correlation between alkenes and $R_L$
or air mass transport time. Therefore, the variability of alkenes in the coastal



atmosphere seems to be weakly impacted by the terrestrial sources when compared to
alkanes. We attribute this to two main factors. First, the mean concentration of alkenes
in the urban air was only 1.4 times of that in marine air, whereas it was 5.4 times for
alkanes. Alkenes undergo more rapid oxidation due to their higher reactivities compared
to alkanes during air mass transport. Secondly, oceanic ventilation may play a more
substantial role in affecting marine alkenes (discussed in section 3.4).

## 3.4 Oceanic impact on marine atmospheric NMHCs composition

**Sea-to-air fluxes of NMHCs**

The mean (range) of sea-to-air fluxes of ethane, propane, i-butane, n-butane, ethylene,
propylene, and isoprene was 44.6 (0.2-118), 41.5 (0.2-157), 31.7 (0.1-146), 10.9 (-0.8-
96.1), 321 (1.7-775), 56.1 (0.2-212), and 112 (0.5-468) nmol m$^{-2}$ d$^{-1}$, respectively, in
the Yellow Sea and the East China Sea (Table 1). These values were comparable to
those reported in Chinese marginal seas (Wu et al., 2021; Li et al., 2021) and 23-38°N
Atlantic Ocean (Tran et al., 2013), but were larger than those reported values in the
North Sea (Broadgate et al., 1997) and the Northwest Pacific Ocean (Li et al., 2019;
Wu et al., 2023) (Table S10).
The mean of sea-to-air fluxes of the total observed NMHCs was 698 ± 607 nmol m$^{-}$
$^{2}$ d$^{-1}$ in areas within 100 km from the coastline, which was relatively higher than that in
the regions beyond 100 km (480 ± 481 nmol m$^{-2}$ d$^{-1}$). These elevated fluxes in the sea
areas closer to land could be attributed to the influence of phytoplankton biomass and
chromophoric dissolved organic matter (CDOM). Seawater NMHCs are not only
directly synthesized by phytoplankton (Ratte et al., 1995), but they can also be emitted
through the photochemical degradation of CDOM (Ratte et al., 1993; Lee and Baker,
1992). To substantiate our findings, we analyzed the monthly Chl-*a* concentration and



the absorption coefficient at 443 nm of seawater in April 2021 from the remote sensing
dataset from the NASA Ocean Color data service (https://oceancolor.gsfc.nasa.gov/)
(Fig. S2). The mean (±SD) of Chl-*a* concentrations was $2.83\pm1.17$ and $1.68\pm1.44$ μg
$L^{-1}$ in the areas within and beyond 100 km from coastline, respectively. Correspondingly,
the mean (±SD) of seawater absorption coefficients at 443 nm was at $0.124\pm0.060$ and
$0.069\pm0.040$ $m^{-1}$, respectively. Hence, the heightened phytoplankton biomass and
enriched photoreaction substrate collectively enhanced both the biological production
and abiotic formation of NMHCs, consequently resulting in a pronounced NMHCs
emission in nearshore regions.
**Assessing the effect of oceanic emission on NMHCs**
Prior to delving into the correlation between oceanic emissions and NMHCs
concentrations, it is imperative to acknowledge the influence of different gases'
reactivity on this relationship. For instance, ethane possesses an atmospheric lifetime
of approximately 78 d at 24 h [•OH] concentration of $6\times10^5$ molecules $cm^{-3}$. This means
all ethane emitted from the ocean within this period to potentially contribute to the
accumulation of atmospheric ethane. Conversely, isoprene, with a much shorter lifetime
of only 0.2 d, emitted within a very brief window can impact its atmospheric level. Thus,
to mitigate the impact of varying reactivity among the different gas species, we
calculated the life-weighted concentrations of each NMHCs according to their
atmospheric lifetime (introduced in section 2.5). This novel method is more nuanced to
assess the impact of oceanic emission on atmospheric NMHCs, as it acknowledging not
only their abundance but also their residence in the atmosphere.
In spite of the elevated oceanic emission of NMHCs within the 100 km from land,
its impact on atmospheric NMHCs composition was comparatively weak displaying a





slope of 0.0187 (Fig. 5c), which was lower than the fitted result of the dataset in areas
beyond 100 km from land with a slope of 0.0415 (Fig. 5d). This could be attributed to
the disturbance of terrestrial outflow in nearshore areas, mitigating the direct impact of
oceanic emission on NMHCs. As it extended further from the land, the terrestrial
influence diminished. This, in turn, strengthens the regulatory impact of oceanic
emission on atmospheric NMHCs levels.

In addition, the average flux of total alkenes across the entire region was $163 \pm 221$

nmol m$^{-2}$ d$^{-1}$, which was approximately 5 times higher than that of alkanes ($32.2 \pm 37.5$
nmol m$^{-2}$ d$^{-1}$). This substantial discrepancy indicates that alkanes and alkenes are
certainly influenced differently by oceanic emissions. The correlation between the
lifetime-weighted concentrations of alkenes and their fluxes was statistically significant
($r = 0.548$, $n = 57$, $p < 0.001$), while it was insignificant for alkanes ($r = 0.113$, $n = 76$,
$p = 0.329$). When specific species of alkanes (Fig. 5e) and alkenes (Fig. 5f) were
separately plotted against their sea-to-air fluxes, alkenes exhibited a steeper slope of
0.0072 compared to the slope of 0.0044 for alkanes. This signifies that oceanic emission
has a more significant impact on atmospheric alkenes compared to alkanes, which
verifies our hypothesis as stated at the end of section 3.3.

## 3.5 Identification and apportionment of the sources of marine atmospheric NMHCs



**Source identification**

Since the chemical compositions are largely controlled by the sources of emissions,

specific ratios of hydrocarbons have been widely employed to identify the sources of
NMHCs (Gilman et al., 2013; Rossabi and Helmig, 2018). For instance, elevated iso-
pentane/n-pentane ratios are indicative of the heavy influence of vehicular emissions





(2.2-3.8) and gasoline fuel evaporation (1.8-4.6) (Gentner et al., 2009; Jobson et al.,
2004; Liu et al., 2008; Russo et al., 2010). Conversely, the lower ratios indicate the
importance of tropical forest fires (0.43-0.57) (Andreae and Merlet, 2001; Rossabi and
Helmig, 2018), natural and oil gas operations (0.81-1.1) (Gilman et al., 2013; Swarthout
et al., 2013), and marine vessel exhaust (1.59-1.71) (Bourtsoukidis et al., 2019) in
controlling the chemical composition of NHMCs. In this study, a significant correlation
was observed between i-pentane and n-pentane (r = 0.67, $p < 0.01$) (Fig. 6), and the i-
pentane/n-pentane ratio spans a wider range from 0.89 to 2.46, suggesting that the
composition of NMHCs in the marginal seas of China is controlled by multiple sources
e.g., natural and oil gas operations, marine vessel exhaust, vehicular emissions, and
gasoline evaporation.
Furthermore, propane, i-butane, and n-butane exhibited strong intercorrelations (r =
0.52-0.95, $p < 0.05$). They also displayed strong correlations with ethane, i-pentane,
and n-pentane (r = 0.55-0.98, $p < 0.05$). These alkanes were recognized as the primary
components of liquid petroleum gases (Blake and Rowland, 1995), extensively utilized
as fuel in taxis, private cars, and public buses in China (Guo et al., 2017; Zhang et al.,
2015). Notably, C3-C5 alkanes also exhibited significant correlations with ethane (r =
0.55-0.72, $p < 0.05$) and carbon monoxide (r = 0.59-0.81, $p < 0.05$), while ethane and
carbon monoxide are acknowledged tracers for fossil fuel or biomass/biofuel
combustion and incomplete combustion, respectively (Lai et al., 2010; Tang et al., 2009;
Parrish et al., 2009). This indicated the contribution of vehicular emissions of liquid
petroleum gases and combustion of fossil fuel or biomass to light alkanes. Additionally,
strong correlations were observed among monocyclic aromatics (benzene, toluene,
ethylbenzene) (r = 0.67-0.83, $p < 0.05$). This finding was consistent with recent
emission inventory research identifying monocyclic aromatics as significant



constituents of ship exhaust (Xiao et al., 2018b; Wu et al., 2019). As for oceanic
emissions, we have presented the sea-to-air fluxes of NMHCs and discussed the
significant effect of oceanic emissions on NMHCs in Section 3.4. Multiple studies
highlighted that the ocean is one of the important sources of these gases (Kato et al.,
2007; Li et al., 2019; Mallik et al., 2013; Sahu et al., 2010; Rudolph and Johnen, 1990).
**Source apportionment**
The potential sources of the atmospheric NMHCs and their respective contributions
to each category were determined using the PMF model. Four isolate factors were
extracted according to their composition profiles depicted in Fig. 7a. These factors,
including industrial production, exhaust emission, terrestrial vegetation, and oceanic
ventilation, were identified based on chemical profiles in literature.
Propane, i-butane, n-butane, i-pentane, n-pentane, and CFC-11 showed strong
loadings (> 70 %) on factor 1. The presence of propane, butanes, and pentanes suggests
the influence of the refinery activities (Buzcu and Fraser, 2006). Additionally, propane
has been recognized as a characteristic NMHCs derived from natural gas emissions and
butane is indicative of liquefied petroleum gas (LPG) (Guo et al., 2011; Tsai et al., 2006;
Hui et al., 2018; Ho et al., 2009). Moreover, CFC-11 is a typical artificial industrial
product. Subsequently, factor 1 was identified as a factor relating to industrial activities.
The profile of factor 2 showed strong loadings of benzene (72 %), toluene (57 %),
and ethylbenzene (64 %), along with moderate impacts of ethylene (34 %) and
propylene (32 %). Benzene emissions are notably associated with vehicle exhaust
(Zhang et al., 2013; Zhang et al., 2016) and considerable fractions of aromatics can be
emitted from ship exhaust during both berthing and cruising (Cooper, 2005; Xiao et al.,
2018a). C2-C4 alkenes could stem from ship emissions in the open ocean (Eyring et al.,





2005). Therefore, factor 2 can be potentially assigned as a source of the exhaust
emissions of vehicles and ships.

Factor 3 was assigned as oceanic ventilation due to elevated percentages of DMS

(74 %) and $CHBr_3$ (53 %), considering the dominant contributions of ocean emission
to DMS (Lana et al., 2011; Lee and Brimblecombe, 2016) and $CHBr_3$ (Quack and
Wallace, 2003; Ashfold et al., 2014). Factor 4 was mainly characterized by a high
percentage of isoprene (68 %), an indicator of biogenic emission from terrestrial
vegetation (Guenther et al., 2006; Wu et al., 2016). However, given isoprene's high
reactivity, this factor should be treated cautiously and regarded as a lower limit (Fujita,
2001). Although its short atmospheric lifetime hinders long-range transport, the
minimum air mass transport time from land to the oceanic station was four hours in this
study, implying the potential for terrestrial isoprene to reach the nearshore atmosphere.

According to the results of the PMF model analysis, the dominant source of

atmospheric alkanes in the Chinese marginal seas was industrial activities (0.253 ppb,
60.8 %), followed by exhaust emissions (0.095 ppb, 23 %). Contributions from
terrestrial vegetation emission (0.049 ppb, 11 %) and oceanic ventilation (0.021 ppb,
5.2 %) were relatively smaller. Furthermore, exhaust emissions (0.017 ppb, 32.5 %),
industrial activities (0.017 ppb, 31 %), and ocean ventilation (0.016 ppb, 30.4 %)
contribute almost equally to atmospheric alkenes. Collectively, these three factors
constitute the main sources of alkenes (93.8 %), whereas the contribution from
terrestrial vegetation is minimal, at merely 6.2 %. Particularly, the contribution of
terrestrial sources to alkanes (89 %) is greater than that to alkenes (69.6 %), while the
contribution of ocean emission to alkenes (30.4 %) is greater than that to alkanes
(5.2 %). This is consistent with the conclusions in section 3.3 and section 3.4.

To assess the environmental implications of different sources, the ozone formation

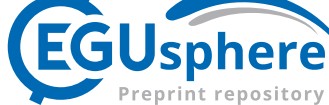



potential (OFP) of NMHCs was calculated using $OFP = MIR \times C$, where $MIR$ depicts
the maximum incremental reactivity and $C$ represents the concentration of NMHCs
(Carter, 1994). Specific data can be seen in supplementary Table S11. The contributions
of different factors to the OFP are as follows: industrial activities (2.30 µg m$^{-3}$, 56 %),
exhaust emissions (0.87 µg m$^{-3}$, 21 %), oceanic ventilation (0.64 µg m$^{-3}$, 16 %), and
terrestrial vegetation emissions (0.27 µg m$^{-3}$, 7 %). Notably, terrestrial sources
collectively accounted for 89 % of alkanes and 69.6 % of alkenes within the coastal
marine atmosphere of China. Furthermore, these terrestrial factors contributed 84 % of
the OFP associated with C2-C5 NMHCs. These findings highlight that terrestrial
outflow substantially constitutes the atmospheric NMHCs and plays a significant role
in regulating air quality in nearshore environments.

## 4 Conclusions


Our study characterized the atmospheric NMHCs in both coastal cities and Chinese
marginal seas, and determined that both oceanic ventilation and terrestrial inputs play
important roles in controlling the distribution and chemical composition of NMHCs in
the coastal atmosphere of China.
Alkanes were the dominant NMHCs both in urban and nearshore atmosphere, and
the atmospheric concentrations of alkanes were significantly higher in coastal cities
compared to coastal seas, showing the potential of terrestrial alkanes as a source of
alkanes in the marine atmosphere through transport. Generally, alkane concentrations
tended to be higher in cases: sampling sites closer to land, longer retention of air mass
over land, and shorter air mass transport time from land to sampling site. However,
these effects could not apply to alkenes due to their higher reactivities and the
substantial sea-to-air fluxes. Additionally, the impact of oceanic emissions on NMHCs



composition was more pronounced in areas beyond 100 km from land compared to
areas within 100 km, because the terrestrial input gradually diminishes along the
direction towards the open ocean.
Combining the outcomes of the PMF model and chemical profiles of diverse sources
in the literature, we extracted four isolated sources of NMHCs in the nearshore
atmosphere. Terrestrial sources (including industrial activities, vehicular exhaust, and
vegetation emission) primarily constitute the NMHCs in the nearshore atmospheres,
further contributing 84 % to the OFP associated with C2-C5 NMHCs. This highlights
the significant influence of terrestrial outflow on the distribution and composition of
NMHCs in the nearshore atmosphere of China, emphasizing the necessity for a
comprehensive understanding of both natural and anthropogenic emissions of NMHCs.
**Code and data availability**
Data presented in this paper are publicly available at Figshare via
https://doi.org/10.6084/m9.figshare.24722286. The remote-sensing datasets of Chl-*a*
and total absorption at 443 nm are available at https://oceancolor.gsfc.nasa.gov. Code
to calculate retention of air mass over land can be download from
https://doi.org/10.1029/2021JD034960 (Zhou et al., 2021).
**Competing interests**
The authors declare that they have no conflict of interest.



## Author contributions


Honghai Zhang and Jian Wang designed the investigation and experiments. Jian
Wang, Qianyao Ma, Feng Xu, Gaobin Xu, Shibo Yan, Jiawei Zhang, and Jianlong Li
collected and determined the samples. Jian Wang analyzed the data and wrote the
manuscript. Honghai Zhang, Lei Xue, Zhaohui Chen, and Guiling Zhang reviewed and
revised the manuscript.

## Acknowledgments


We thank the chief scientist, captain, and crews of the R/V '*Dong Fang Hong 3*' for
assistance and cooperation during the investigation. We would like to acknowledge the
NOAA Air Resource Laboratory for the provision of the HYSPLIT trajectory model
used in this study and the NASA Ocean Color data service for provision of the remote-
sensing dataset of Chl-*a* and total absorption at 443 nm in this study region.

## Financial support


This work was financially supported by the National Natural Science Foundation of
China (42276042, 41876082, and 42006044); the Laoshan Laboratory (LSKJ
202201701), the Fundamental Research Funds for the Central Universities (202372001
and 202072001).

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



## Figure Captions

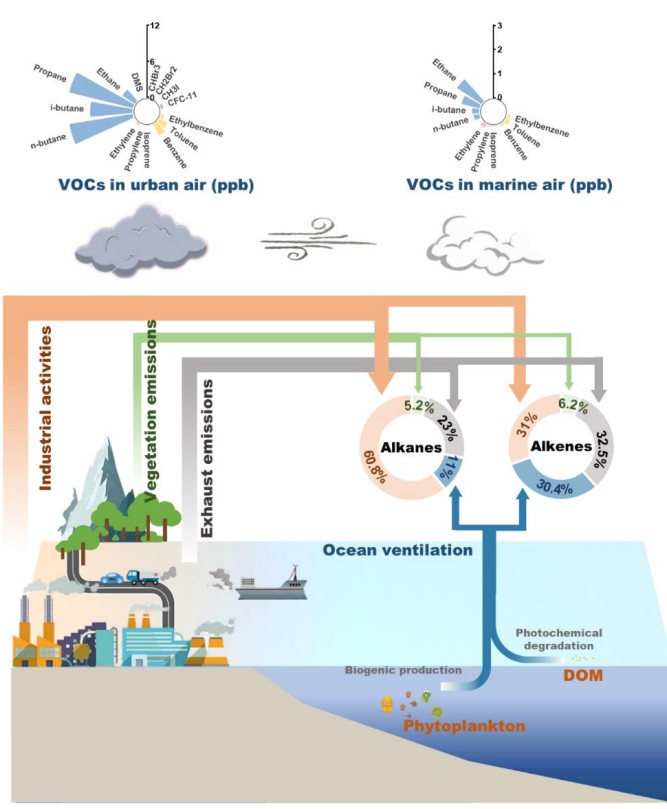

**Graphical abstract** Schematic diagram showing the main sources and their relative contributions to the non-methane hydrocarbons (NMHCs) budget in the nearshore atmospheres of China. The ring bar chart above the land or the ocean shows the composition of urban or marine atmospheric trace gases determined in this study. The axes with unit of ppb indicate the atmospheric concentrations of gases. The distinct colored wedges indicate the alkanes (skyblue), alkenes (pink), monocyclic aromatics (yellow), volatile halogenated compounds (VHCs, lilac), and dimethyl sulfur (DMS, palegreen). Note that only alkanes, alkenes, and monocyclic aromatics are shown in marine atmosphere. The colored arrows or annuli indicate the main sources of NMHCs in offshore atmosphere: industrial activities (sandybrown), exhaust emissions (darkgray), oceanic ventilation (steelblue), and vegetation emissions (lightgreen). The numbers on the annuli are their respective relative contributions.



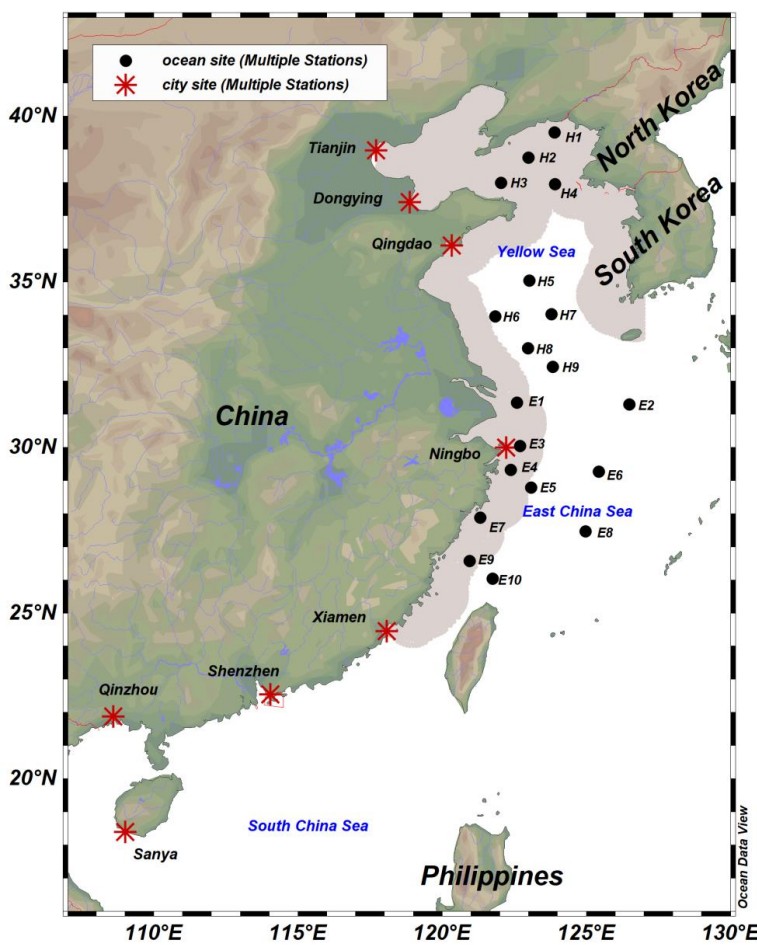


**Figure 1** Map showing the sampling stations in the coastal cities (red asterisks) and marginal seas

(black dots) of China from March to May 2021. The gray shaded area represents the inshore region

within 100 km from the coastline.

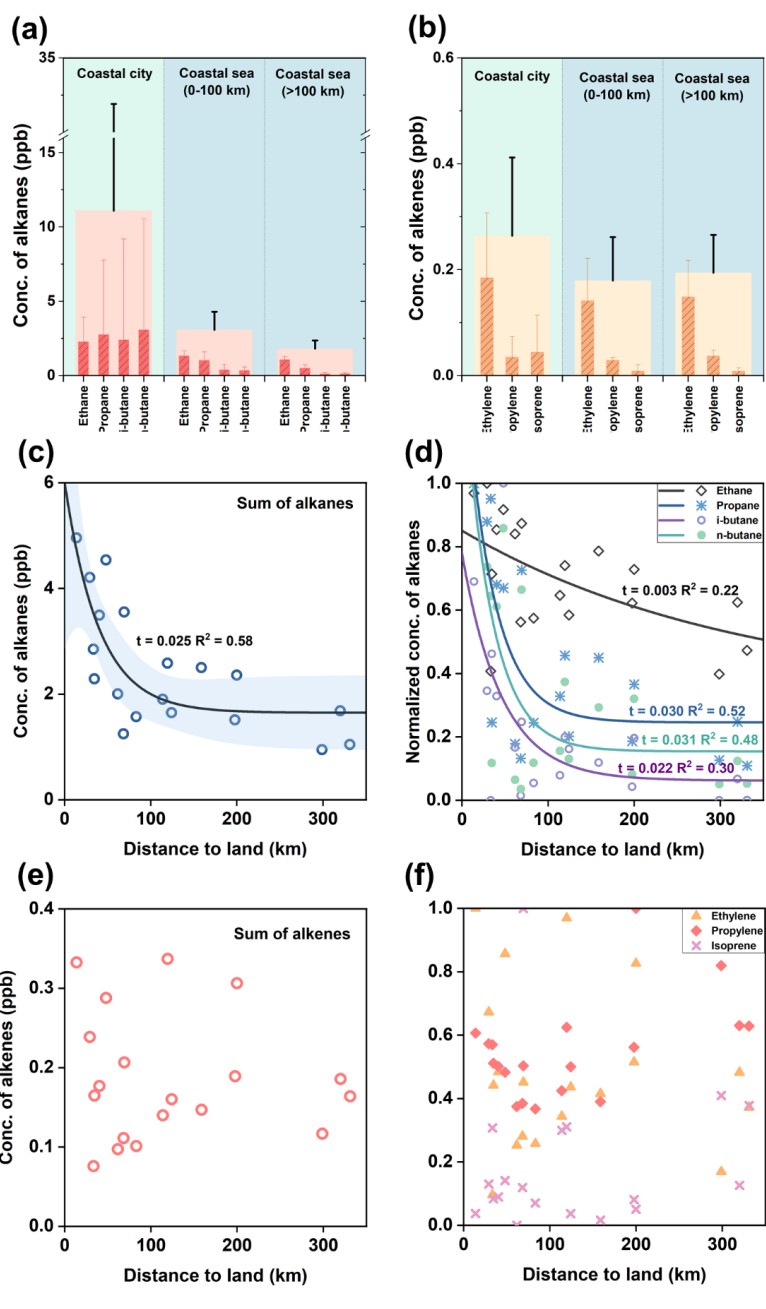


**Figure 2** Means of the concentrations of alkanes (panel a) and alkenes (panel b) in the atmosphere

over coastal cities (n = 14) and nearshore (0-100 km, n = 10) and offshore (>100 km, n = 9) coastal



seas of China. The wider columns in panel a or b represent the sums of individual alkanes or alkenes
with error bars depicting the propagated errors from each NMHCs. Summed alkane (panel c) or
alkene (panel e) and normalized concentrations of specific alkane (panel d) or alkene (panel f)
plotted as a function ($y = Ae^{-tx}+y_0$) of the distance from sampling sites to the nearest land.

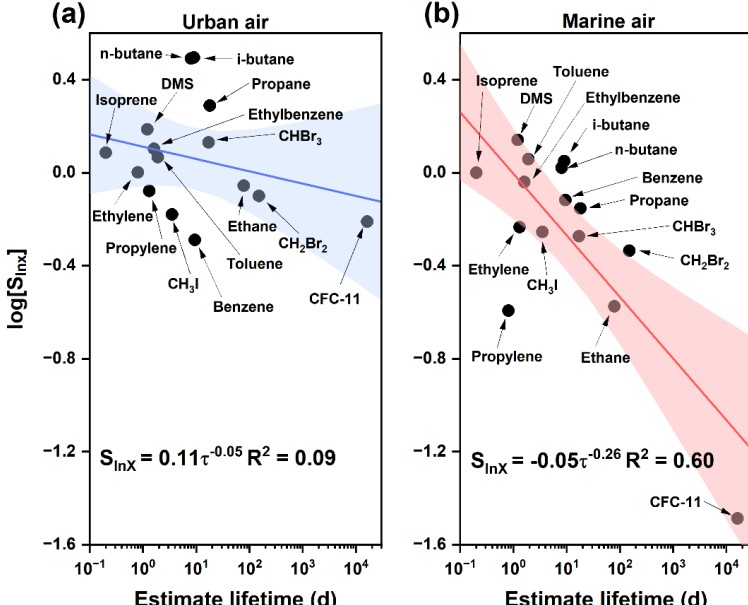

**Figure 3** Atmospheric variability ($\log[S_{lnx}]$) plotted as a function of the estimated •OH lifetime for
each non-methane hydrocarbons (NMHCs) from the coastal cities (panel a) and marginal seas of
China (panel b). The blue or red line is the best linear fitting. Shadowed area represents the
confidence band at a 95 % confidence level.



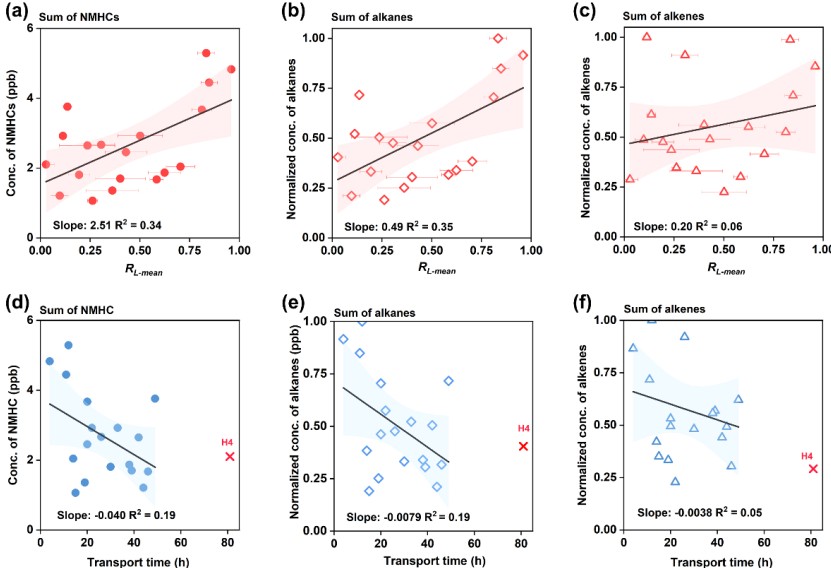


**Figure 4** Concentrations of non-methane hydrocarbons (NMHCs) combined (panel a or d), alkanes

(panel b or e), and alkenes (panel c or f) at each site plotted against the mean retentions of air mass

over land ($R_{L-mean}$, n = 3) or the transport time of air mass, respectively. The error bars for

$R_{L-mean}$ indicate the standard deviation from three different time-scale trajectories (48h, 72h, and

96h). The black line is the best fitting of liner function and shadowed area represents the confidence

band at a 95 % confidence level. H4 (marked with red "×") is treated as an outlier since it alone

deviates from the main dataset.

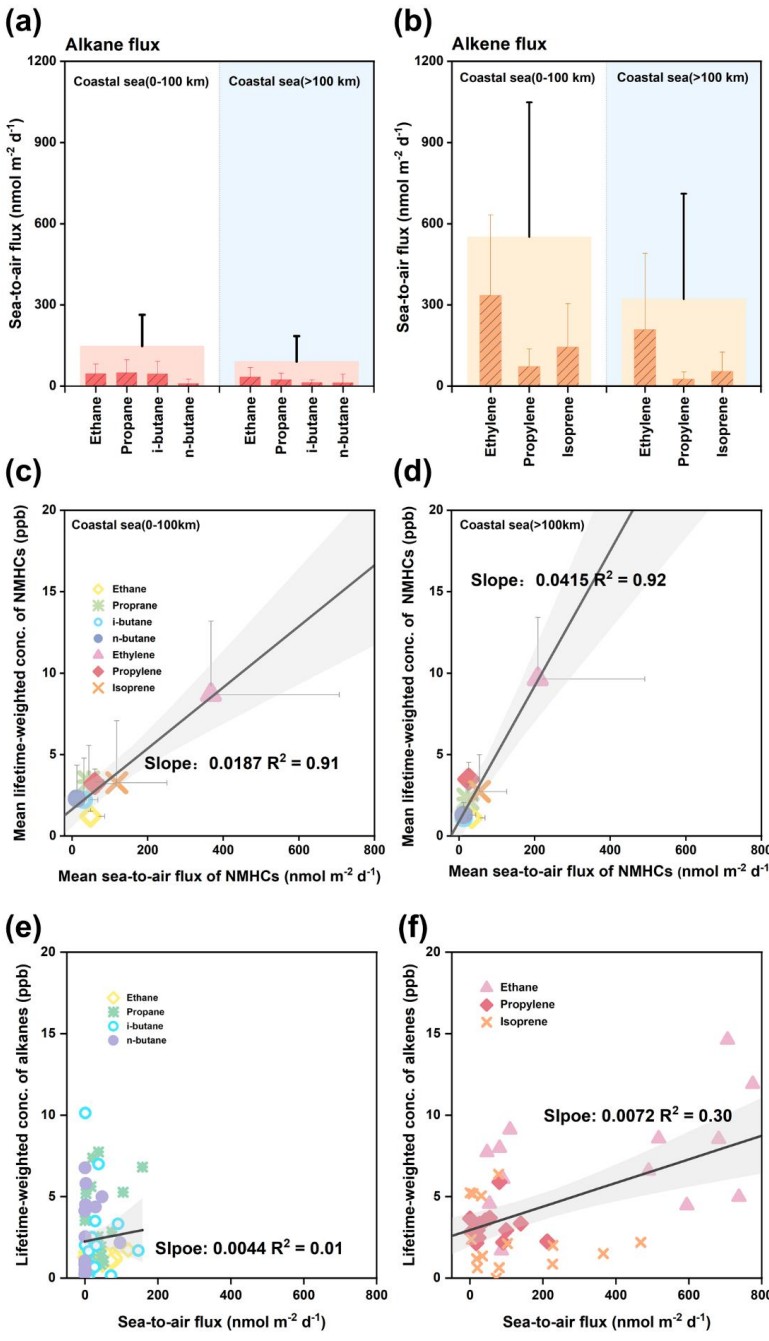

Figure 5 Means of sea-to-air fluxes of alkanes (panel a) and alkenes (panel b) in sea areas within

100 km (n = 10) and beyond 100 km (n = 9) from the nearshore land. The wider columns represent



the sum of alkanes or alkenes. Panel c or d shows the means of lifetime-weighted concentrations of
NMHCs plotted against the means of their mean sea-to-air fluxes in the area within 100 km or
beyond 100 km from the coastline. Specific lifetime-weighted concentrations of alkanes (panel e)
and alkenes (panel f) plotted against sea-to-air fluxes in the whole coastal sea region. The black,
blue or red line is the best linear fitting for each dataset and shadowed area represents the confidence
band at a 95 % confidence level.

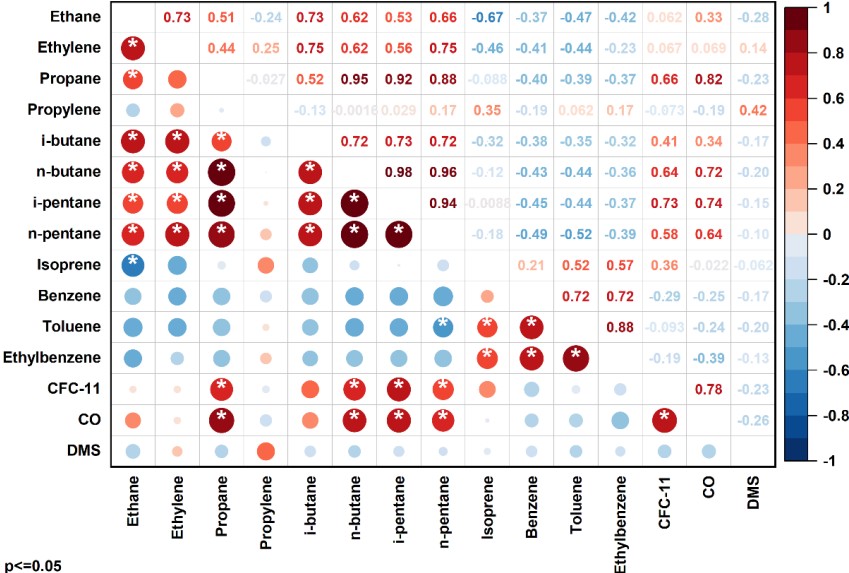


**Figure 6** Correlation coefficients (r) between the various trace gases determined in the atmosphere
over the Yellow Sea and the East China Sea. The white asterisk means the correlation is significant
at the *p*<0.05 level. The color of dots, red or blue, indicates the positive or negative correlation and
the size of the dots indicates the absolute value of r.



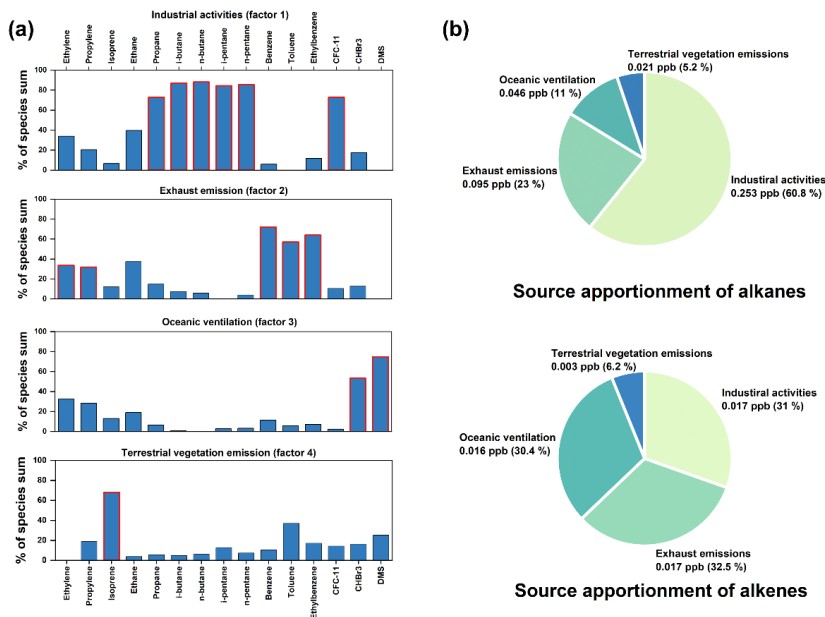


**Figure 7** Representative factor profiles from the positive matrix factorization (PMF) model (panel

a). Non-methane hydrocarbons (NMHCs) marked with red rim are selected as indicators for the

specific factors. Relative contributions of different factors/sources to the alkanes and alkenes in the

oceanic atmosphere (panel b).





**Table**

874

**Table 1** Atmospheric and seawater concentrations, sea-to-air fluxes, and the calculated atmospheric lifetime of each non-methane hydrocarbons (NMHCs) based on the reaction with hydroxyl radicals (•OH).

| Species | Conc. in urban air (ppb) | Conc. in oceanic air (ppb) | Conc. in seawater (pmol L⁻¹) | Sea-to-air flux (nmol m⁻² d⁻¹) | Atmospheric lifetime[b] (d) |
|---|---|---|---|---|---|
| Ethane | 2.26 (0.277-5.72) | 1.24 (0.686-1.72) | 11.6 (4.70-22.8) | 44.6 (0.2-118) | 78 |
| Propane | 2.95 (0.149-20.1) | 0.822 (0.226-1.79) | 12.6 (3.68-136) | 41.5 (0.2-157) | 18 |
| i-butane | 2.57 (BD[a]-27.6) | 0.283 (BD-1.17) | 9.46 (1.54-35.3) | 31.7 (0.1-146) | 9.1 |
| n-butane | 3.29 (0.018-30.2) | 0.256 (0.025-0.694) | 4.95 (BD-32.9) | 10.9 (-0.8-96.1) | 8.2 |
| Ethylene | 0.180 (0.035-0.390) | 0.151 (0.028-0.295) | 70.4 (8.40-136) | 321 (1.7-775) | 2.3 |
| Propylene | 0.036 (BD-0.129) | 0.033 (0.022-0.060) | 15.2 (2.42-27.6) | 56.1 (0.2-212) | 0.73 |
| Isoprene | 0.046 (0.006-0.250) | 0.008 (BD-0.043) | 31.0 (3.43-105) | 112 (0.5-468) | 0.19 |

[a]: Below the detection limit.

[b]: Assuming an average [•OH] of $6\times10^5$ molecules cm⁻³ within 24 h (Jobson et al., 1999), and using the rate constant with •OH at 288 K taken from Atkinson et al. (1997).