# Peer review of "Roles of oceanic ventilation and terrestrial outflow in the atmospheric non-methane hydrocarbons over the Chinese marginal seas"

_EGUsphere, 2023_

## Author Comment (AC1)

**An itemized response (blue words) to the reviewer's comments and suggestions**

We sincerely appreciate the reviewers for the comments concerning our manuscript entitled "Roles of oceanic ventilation and terrestrial outflow in the atmospheric non-methane hydrocarbons over the Chinese marginal seas" [EGUSPHERE-2023-2935]. These comments are all valuable and very helpful for improving our paper and have an important guiding significance to our future research. We have made changes to the manuscript and tried our best to improve the manuscript following these comments. The changed portion in the revised manuscript is highlighted in blue. The primary corrections in the revised manuscript and the detailed responses to the comments of the reviewer are as follows.

**Reviewer #1 Evaluations:**

The manuscript is mainly describing the specific effects of ocean emission and terrestrial input on the atmospheric NMHCs in a representative marginal sea. While the impact of terrestrial input on the coastal environment is widely recognized, a more detailed discussion has been lacking. The authors address the influence mechanism of terrestrial land-based inputs on nearshore atmospheric NMHCs, taking into the distance from the shore and relevant parameters of air masses, and assess the relative contributions from land and ocean sources. This is meaningful to deepen the understanding of how terrestrial input affecting the marine environment near the coast.

I think the manuscript is suitable for publication in the ACP after a revision.

**Reply:** We sincerely appreciate the reviewer's valuable comments and suggestions, which help us improve the manuscript. Following the reviewer's comments, we made a revision to the manuscript. All changes in the revised manuscript are highlighted in blue. We also provide below a point-to-point list regarding the changes made in the text for the reviewer's convenience to review.

Major comments or suggestions:

1. The positive matrix factorization model is the methodology employed by the authors to deconvolute the source factors of atmospheric NMHCs and assess their respective contributions. The authors should provide a more detailed exposition on the principles of this approach in Method section to enhance clarity and understanding.

**Reply:** Thanks for your valuable suggestion. Details for the PMF model have been supplied in the manuscript, as indicated below:

Line 221-245:

"PMF model introduced in detail in the study of Paatero and Trapper (1994) was applied to analyze the data of atmospheric NMHCs in the Yellow Sea and the East China Sea. Based on a matrix consisting of the concentrations of diverse chemical species, the objective of PMF is to determine the number of NMHCs source factors, the chemical composition profile of each factor, and the contribution of each factor to species. The matrix representation of this model is as follows:

$$x_{ij} = \sum_{k=1}^{p} g_{ik}f_{kj} + e_{ij} \qquad\qquad\qquad (9)$$

Where $x_{ij}$ represents the concentration of species $j$ measured on sample $i$, $p$ denotes the number of factors facilitating the samples. $f_{kj}$ represents the concentration of species $j$ in factor profile $k$, $g_{ik}$ denotes the relative contribution of factor k to sample

$i$, and $e_{ij}$ represents the PMF model error of species $j$ measured on sample $i$. The factors resolved by PMF are typically interpreted as sources. The objective of this algorithm is to find the values of $f_{kj}$, $g_{ik}$, and $p$ that best reproduce $x_{ij}$, continuously adjusting $f_{kj}$ and $g_{ik}$ until the minimum $Q$ value for a given $p$ is attained. $Q$ is defined as:

$$Q = \sum_{i=1}^{n} \sum_{j=1}^{m} \left(\frac{e_{ij}}{\sigma_{ij}}\right)^2 \qquad\qquad\qquad (10)$$

Where $\sigma_{ij}$ represents the uncertainty of the concentration of the species $j$ in sample $i$,

$n$ is the number of samples, and $m$ is the number of species. In applying the PMF

model, the significance of missing data in the matrix was decreased by using the species median. The uncertainty for normal data was estimated as 20 % of the NMHCs concentrations because the analytical uncertainty was not available (Buzcu and Fraser,

2006). The model ran 20 times and we selected the result with the minimum $Q$ value.

Besides, approximately 94 % of the scaled residuals given by PMF ranged from -3 to 3

(Fig. S1), suggesting a reasonable fit of the model result."

2. While the authors have made a clear statement regarding the significant difference in NMHCs between urban and marine atmosphere, there is still a need for a more explicit expression on the distribution characteristics of NMHCs, especially in the marine atmosphere. It is advisable to make advancements in the description within the main text, ideally supplemented with relevant Figure for a comprehensive view.

**Reply:** Thanks for your valuable suggestion. We have represented an explicit description about the distribution of NMHCs in the marine atmosphere in the main text and provided relevant Figures in supporting information, as indicated below:

Line 271-277:

"In spatial terms, multiple NMHCs (e.g. ethane, propane, i-butane, n-butane, and ethylene) showed higher atmospheric concentrations in regions closer to the land. The elevated concentrations are primarily concentrated along the coastal regions of the East China Sea and the north Yellow Sea (Figure S3). The disparity in NMHCs concentrations between land and ocean, as well as the distribution pattern of NMHCs in the marine atmosphere, suggested the potential influence of terrestrial sources on the oceanic NMHCs."

[Figure]

**Figure S3** Distributions of alkanes (black dots) and alkenes (red dots) in the atmosphere over the Yellow Sea and the East China Sea

3. Obviously, the distance from the land to the oceanic station is a crucial parameter in the authors' discussion. However, I didn't find any information about the source of distance data or an introduction to the relevant calculation method. The authors should explicitly provide details on these aspects for clarity and transparency in the manuscript.

**Reply:** We are sorry for our negligence about the introduction of the calculation method for the distance from the land to the oceanic stations. It has been supplied in the manuscript, as indicated below:

**Line 196-202:**

"**2.7 Calculation of the shortest distance from the sampling station to the land**

Coastline latitude and longitude data near the study area (20-45°N, 110-130°E) were extracted from the World Vector Shorelines (downloaded from https://www.ngdc.noaa.gov/mgg/shorelines/data/gshhg/latest/). Subsequently, distances from the maritime sampling stations to all coastal locations were computed. The minimum value among these distances was selected as the shortest distance to the land (listed in Table S9)."

4. The authors have employed an innovative approach in assessing the impact of sea-air flux on the marine NMHCs, by calculating the atmospheric lifetime-weighted concentrations of different gases. In this way, the variabilities of atmospheric reactivities of different gases were considered simultaneously when examining the relationship between flux and concentration. It is found intriguing and seemed effective. Furthermore, the authors could extend this novel idea to the discussion about the air mass. Both of land retention and transport time of air masses serve as indicators of air mass characteristics, reflecting the impact of land-based inputs. Combining these two parameters to collectively explore the impact of air masses on NMHCs might offer a new perspective, potentially leading to fresh discoveries

**Reply:** Thanks for your valuable suggestion. As combining the effects of $R_L$ and transport time of air mass on marine NMHCs, it emphasized the possibility of the terrestrial influence on the marine atmospheric environment.

Line 343-347:

"Notably, elevated alkane concentrations were affected by those air masses with larger $R_{L-mean}$ (>0.8) and shorter transport time (<20 h) (Figure S4). This emphasized the terrestrial influence on alkanes in the marine atmosphere, since both $R_L$ and transport time serve as indicators of air mass terrestrial characteristics."

[Figure]

**Figure S4** Impacts of air mass (indicated by transport time and $R_L$) on atmospheric alkanes (a) and alkenes (b) over the Yellow Sea and the East China Sea

**Minor comments:**

1. Show the standard deviation when you mentioned average, like line 266 "the mean (range) concentration of ethane, propane, i-butane, and n-butane was 2.26 (0.277-5.72), 2.95 (0.149-20.1), 2.57 (BD-27.6), and……"

**Reply:** Thanks for your valuable suggestion. It has been revised in the manuscript, as indicated below:

Line 251-254:

"In urban atmosphere (n = 14), the mean (range) concentration of ethane, propane, i-butane, and n-butane was 2.26 ± 1.66 (0.277-5.72), 2.95 ± 5.12 (0.149-20.1), 2.57 ± 6.99 (BD-27.6), and 3.29 ± 7.68 (0.018-30.2) ppb, respectively (Table 1)."

Line 257-260:

"For alkene species in the urban atmosphere (n = 14), the mean (range) of ethylene, propylene, and isoprene was 0.180 ± 0.126 (0.035-0.390), 0.036 ± 0.040 (BD-0.129), and 0.046 ± 0.072 (0.006-0.250) ppb, respectively."

Line 262-266:

"In the marine atmosphere (n = 19), the mean (range) concentration of ethane, propane, i-butane, n-butane, ethylene, propylene, and isoprene was 1.24 ± 0.298 (0.686-1.72), 0.822 ± 0.518 (0.226-1.79), 0.283 ± 0.302 (BD-1.17), 0.256 ± 0.214 (0.025-0.694), 0.151 ± 0.077 (0.028-0.295), 0.033 ± 0.009 (0.022-0.060), and 0.008 ± 0.010 (BD-0.043) ppb, respectively."

Line 366-370:

"The mean (range) of sea-to-air fluxes of ethane, propane, i-butane, n-butane, ethylene, propylene, and isoprene was 44.6 ± 35.0 (0.2-118), 41.5 ± 39.9 (0.2-157), 31.7 ± 38.2 (0.1-146), 10.9 ± 25.4 (-0.8-96.1), 321 ± 294 (1.7-775), 56.1 ± 55.2 (0.2-212), and 112 ± 134 (0.5-468) nmol $m^{-2}$ $d^{-1}$, respectively, in the Yellow Sea and the East China Sea (Table 1)."

2. line 358 "ethane possesses an atmospheric lifetime of approximately 78 d at 24 h

[•OH] concentration of 6×105 molecules cm-3……". References are needed to illustrate the source and credibility of the data used here.

**Reply:** Thanks for your valuable suggestion. We have cited the relevant references as indicated below:

"For instance, ethane possesses an atmospheric lifetime of approximately 78 d at 24 h

[•OH] concentration of $6×10^5$ molecules cm$^{-3}$ (Jobson et al., 1999), using the rate constant with •OH at 288 K taken from Atkinson et al. (1997)."

3 line 468 "ozone formation potential (OFP) of NMHCs was calculated using OFP =

MIR × C…" The calculation description should be in the method section and clearly present the specific constants used in the equation and their literature sources.

**Reply:** Thanks for your valuable suggestion. The calculation description has been composed in the method section 2.5 and the specific constants have been presented in

Table S11.

**"5 Calculation of ozone formation potential of NMHCs**

$$OFP_i = MIR_i \times C_i \tag{5}$$

To assess the environmental implications of different sources, the ozone formation potential (OFP) of NMHCs was calculated using Eq. (5), where $MIR_i$ depicts the maximum incremental reactivity and $C_i$ represents the concentration of NMHCs (Carter, 1994). Specific data was listed in supplementary Table S11."

4 Is it necessary to include both the full term and abbreviation 'NMHCs' in the caption of each Figure? Generally, after the initial mention in the text, subsequent references can use the abbreviation alone for conciseness.

**Reply:** Thanks for your valuable suggestion. After the first appearance of the full name, we use abbreviations in the following text.

5 Increase the font size of the text in the Figures to make them more readable.

**Reply:** We feel sorry for the blurred figures due to the inappropriate font size. We have increased the font size of the text in the Figures, e.g. Figure 7.

[Figure]

**Figure 7** Representative factor profiles from the PMF model (panel a) and relative contributions of different factors/sources to the alkanes and alkenes in the oceanic atmosphere (panel b). NMHCs in panel a marked with red rim are selected as indicators for the specific factors.

---

## Author Comment (AC3)

**An itemized response (blue words) to the reviewer's comments and suggestions**

We sincerely appreciate the reviewer for the comments concerning our manuscript entitled "Roles of oceanic ventilation and terrestrial outflow in the atmospheric non-methane hydrocarbons over the Chinese marginal seas" [EGUSPHERE-2023-2935]. These comments are all valuable and very helpful for improving our paper and have an important guiding significance to our future research. We have made changes to the manuscript and tried our best to improve the manuscript following these comments. The changed portion in the revised manuscript is highlighted in blue. The primary corrections in the revised manuscript and the detailed responses to the comments of the reviewer are as follows.

This paper presents an analysis of canister measurements of NMHCs at several coastal Chinese cities and over the Chinese marginal seas (along with seawater measurements) to assess the relative impact of oceanic versus terrestrial sources over the seas. Their analysis, which used back trajectories and air-sea fluxes calculated from their measurements, determined that alkanes were primarily impacted by terrestrial sources, whereas alkenes had a larger contribution from oceanic ventilation, with higher NMHC ocean fluxes closer to the coast. PMF analysis confirmed these findings, and provided a more detailed source apportionment of the contributions from industrial, vehicle, terrestrial, and oceanic sources.

I found this paper to be relatively well-written and provides new data in a region that has not been frequently studied. However, as is the conclusions are underwhelming; the fact that this study "highlights the significant influence of terrestrial outflow on the distribution and composition of NMHCs in the nearshore atmosphere of

China…"doesn't seem to me a surprising or new finding. I think the paper needs to do more to demonstrate the implications of this—it begins to do so in lines 467-478 with the discussion of ozone formation potential, but I think it needs more. I suggest editing down some of the correlation and distance analysis earlier in the paper (which all reiterates the same conclusions) and expanding the final section. Perhaps the authors could include a modeling analysis on the impact of air quality (ozone and SOA) over the marginal seas? Or, they could at least expand the source apportionment section and do more to highlight what are the underlying air quality implications? I would support publication after this main concern has been addressed.

**Reply:** Thanks for your suggestion. In order to assess the impact of air quality (ozone and SOA) over the marginal seas, we calculated the ozone formation potential (OFP) and secondary organic aerosol formation potential ($P_{SOAP}$) of atmospheric C2-C5 NMHCs. In addition, we examined the impact of terrestrial outflow on oceanic atmosphere environment, in conjunction with the tropospheric aerosol concentrations and ozone level during the investigation period. The modifications are as follows:

**Line 175-185:**
**"2.5 Calculation of OFP and $P_{SOAP}$ of NMHCs**
To assess the environmental implications of NMHCs, the ozone formation potential ($OFP$, µg m$^{-3}$) and secondary organic aerosol (SOA) formation potential ($P_{SOAP}$, µg m$^{-3}$) are calculated using Eq. (5) and Eq. (6), respectively (Carter, 1994).

$$OFP_i = MIR_i \times C_i \tag{5}$$

$$P_{SOAPi} = \sum C_i \times SOAP_i \times FAC_{toluene}/100 \tag{6}$$

[revised manuscript text omitted]

**Specific comments**

Line 57-60: There are several more recent studies that provide higher/different global VOC ocean emission estimates than Guenther et al. (1995). Suggest providing a range here to expand the literature review and also highlight how uncertain these fluxes here.

**Reply:** Thanks for your suggestion. We have updated the range of VOCs emissions and highlighted their uncertainties, as indicated below:

**Line 57-61:**

"Despite the uncertainties in the global flux of VOCs, substantial evidence indicates a significant discrepancy between terrestrial emissions (660-1146 Tg C yr$^{-1}$) (Guenther et al., 1995, 2012; Messina et al., 2016; Sindelarova et al., 2014; Singh and Zimmerman, 1992) and marine emissions (5-36 Tg C yr$^{-1}$) (Guenther et al., 1995; Singh and Zimmerman, 1992)."

**Reference:**

Guenther, A., Hewitt, C. N., Erickson, D., Fall, R., Geron, C., Graedel, T., Harley, P., Klinger, L., Lerdau, M., McKay, W. A., Pierce, T., Scholes, B., Steinbrecher, R., Tallamraju, R., Taylor, J., and Zimmerman, P.: A global-model of natural volatile organic-compound emissions, J. Geophys. Res.-Atmos., 100, 8873-8892, https://doi.org/10.1029/94JD02950, 1995.

Guenther, A. B., Jiang, X., Heald, C. L., Sakulyanontvittaya, T., Duhl, T., Emmons, L. K., and Wang, X.: The Model of Emissions of Gases and Aerosols from Nature version 2.1 (MEGAN2.1): an extended and updated framework for modeling biogenic emissions, Geosci. Model Dev., 5, 1471–1492, https://doi.org/10.5194/gmd-5-1471-2012, 2012.

Messina, P., Lathière, J., Sindelarova, K., Vuichard, N., Granier, C., Ghattas, J., Cozic, A., and Hauglustaine, D. A.: Global biogenic volatile organic compound emissions in the ORCHIDEE and MEGAN models and sensitivity to key parameters, Atmos. Chem. Phys., 16, 14169–14202, https://doi.org/10.5194/acp-16-14169-2016, 2016.

Sindelarova, K., Granier, C., Bouarar, I., Guenther, A., Tilmes, S., Stavrakou, T., Müller, J.-F., Kuhn, U., Stefani, P., and Knorr, W.: Global data set of biogenic VOC emissions calculated by the

MEGAN model over the last 30 years, Atmos. Chem. Phys., 14, 9317–9341, https://doi.org/10.5194/acp-14-9317-2014, 2014.

Singh, H. B. and Zimmerman, P.: Atmospheric distributions and sources of non-methane hydrocarbons, Nriagu, J. O. (Ed.), Gaseous Pollutants: Characterisation and Cycling, Wiley, New York, p. 235, 1992.

Section 2.7: More detail is needed here to explain the PMF model set-up and what is meant by the scaled residuals shown in Fig. S1.

**Reply:** Thanks for your suggestion. We have added a more detailed explanation of the PMF model and provided further clarification on the meaning of Fig. S1, as indicated below:

**Line 235-248:**

"PMF model introduced in detail in the study of Paatero and Trapper (1994) was applied to analyze the data of atmospheric NMHCs in the Yellow Sea and the East China Sea. Based on a matrix consisting of the concentrations of diverse chemical species, the objective of PMF is to determine the number of NMHCs source factors, the chemical composition profile of each factor, and the contribution of each factor to species. The matrix representation of this model is defined as Eq. (10).

$$x_{ij} = \sum_{k=1}^{p} g_{ik} f_{kj} + e_{ij} \tag{10}$$

Where $x_{ij}$ represents the concentration of species $j$ measured on sample $i$, $p$ denotes the number of factors facilitating the samples. $f_{kj}$ represents the concentration of species $j$ in factor profile $k$, $g_{ik}$ denotes the relative contribution of factor k to sample $i$, and $e_{ij}$ represents the PMF model error of species $j$ measured on sample $i$. The factors resolved by PMF are typically interpreted as sources. The objective of this algorithm is to find the values of $f_{kj}$, $g_{ik}$, and $p$ that best reproduce $x_{ij}$, continuously adjusting $f_{kj}$ and $g_{ik}$ until the minimum $Q$ value for a given $p$ is attained. $Q$ is defined as Eq. (11).

$$Q = \sum_{i=1}^{n} \sum_{j=1}^{m} \left(\frac{e_{ij}}{\sigma_{ij}}\right)^2 \tag{11}$$

Where $\sigma_{ij}$ represents the uncertainty of the concentration of the species $j$ in sample $i$, $n$ is the number of samples, and $m$ is the number of species."

**Line 253-257:**

"Additionally, scaled residuals are instrumental in assessing the fit of the PMF model to the observed data. They represent the difference between the observed and modeled data, scaled by the uncertainty in the observed data. In this PMF analysis, approximately 94 % of the scaled residuals ranged from -3 to 3 (Fig. S1), suggesting a reasonable fit of the model result."

Lines 338-334: This analysis is interesting—were there also any temperature and/or windspeed differences within and beyond 100 km from shore? Also, could the authors find a way to reference Fig 5a and b here, rather than just listing the total (alkanes+alkenes) emissions in text? It's a nice figure and I find it easier to digest than the numbers in Table 1, but right now I don't think those panels are actually referenced in the document.

**Reply:** Thanks for your suggestion. The temperature/windspeed within and beyond 100 km from the shore has been presented in the manuscript, with a discussion about their effects on NMHCs emissions. Besides, Fig. 5a/b has been referenced in the text, as indicated below:

**Line 385-390:**

"The averaged sea-to-air fluxes of alkanes and alkenes within 100 km from the coastline were $147 \pm 116$ and $551 \pm 497$ nmol m$^{-2}$ d$^{-1}$, respectively, which were relatively higher than those beyond 100 km (Fig. 5a, b). Since there were no significant differences in surface seawater temperature and 10 m wind speed between regions within and beyond

100 km from the coastline (Fig. S5), the discrepancy in fluxes might not be driven by physical processes."

[Figure]

**Figure S5** Comparison in surface seawater temperature (panel a) and wind speed (panel b) between regions within/beyond 100 km from the coastline. Boxes span the interquartile range, with lines at the median. Diamonds indicate mean values, and whiskers span the 5-95 percentiles.

**Technical comments:**

Line 45: suggest deleting the word "second" before "organic"

**Reply:** Thanks for your suggestion. We have deleted the word "second", as indicated below:

**Line 45:**

"Non-methane hydrocarbons (NMHCs), a significant subset of volatile organic compounds (VOCs), are acknowledged as key precursors to tropospheric ozone formation (Houweling et al., 1998; Solomon et al., 2005) and organic aerosol generation (Hallquist et al., 2009; Wu and Xie, 2018), playing a pivotal role in atmospheric chemistry."

Line 108: Tables are being referenced out of order here. Suggest reordering to avoid this.

**Reply:** We apologize for any confusion caused by the tables that are out of order. We have reordered table reference in the text, as indicated below:

**Line 108:**
"Sampling details for urban (Table S4) and marine samples (Table S5) are shown in supplementary tables."

Line 177: "analyzed" should be "analyze"

**Reply:** We apologize for the grammar errors. It has been revised in the text, as indicated below:

**Line 192:**
"A novel approach was employed to analyze the correlation between the concentrations of various NMHCs and their sea-to-air fluxes."

Line 181: "yield" should be "yielding"

**Reply:** We apologize for the grammar errors. It has been revised in the text, as indicated below:

**Line 196:**
"This was achieved by dividing the concentration of each NMHCs by its corresponding atmospheric •OH lifetime, yielding a "lifetime-weighted concentration" for each NMHCs ($C_{life-i}$) (Eq. 6)."

Line 359-361: this sentence is awkward, consider revising.

**Reply**: We apologize for any confusion caused by this awkward sentence. It has been revised in the text, as indicated below:

**Line 411-412:**

"The relatively long atmospheric residence time of ethane facilitates its accumulation in the atmosphere."

Line 364: "life-weighted" should be "lifetime-weighted"

**Reply:** Thanks for your suggestion. It has been revised in the text, as indicated below:

**Line 415:**

"Thus, to mitigate the impact of varying reactivity among the different gas species, we calculated the lifetime-weighted concentrations of each NMHCs according to their atmospheric lifetime (introduced in section 2.5)."

Line 366: "acknowledging" should be "acknowledges"

**Reply:** We apologize for the grammar errors. It has been revised in the text, as indicated below:

**Line 417:**

"This novel method is more nuanced to assess the impact of oceanic emission on atmospheric NMHCs, as it acknowledges not only their abundance but also their residence in the atmosphere."

Line 399: "NHMCs" should be "NMHCs"

**Reply:** We apologize for the spelling errors. It has been revised in the text, as indicated below:

**Line 450:**

"Conversely, the lower ratios indicate the importance of tropical forest fires (0.43-0.57) (Andreae and Merlet, 2001; Rossabi and Helmig, 2018), natural and oil gas operations (0.81-1.1) (Gilman et al., 2013; Swarthout et al., 2013), and marine vessel exhaust (1.59-1.71) (Bourtsoukidis et al., 2019) in controlling the chemical composition of NMHCs"

Figure 5e and f: "Slpoe" should be "Slope" in these two panels

**Reply:** We apologize for the spelling errors. It has been revised in the Figure, as indicated below:

**Line 450:**

"Conversely, the lower ratios indicate the importance of tropical forest fires (0.43-0.57) (Andreae and Merlet, 2001; Rossabi and Helmig, 2018), natural and oil gas operations (0.81-1.1) (Gilman et al., 2013; Swarthout et al., 2013), and marine vessel exhaust (1.59-1.71) (Bourtsoukidis et al., 2019) in controlling the chemical composition of NMHCs"

[Figure]

**Figure 5** Means of sea-to-air fluxes of alkanes (panel a) and alkenes (panel b) in sea areas within 100 km (n = 10) and beyond 100 km (n = 9) from the nearshore land. The wider columns represent the sum of alkanes or alkenes. Panel c or d shows the means of lifetime-weighted concentrations of NMHCs plotted against the means of their mean sea-to-air fluxes in the area within 100 km or beyond 100 km from the coastline. Specific lifetime-weighted concentrations of alkanes (panel e) and alkenes (panel f) plotted against sea-to-air fluxes in the whole coastal sea region. The black, blue or red line is the best linear fitting for each dataset and the shadowed area represents the confidence band at a 95 % confidence level.

Figure S2: Suggest changing the panel titles to eliminate the "@Dummy=first"

**Reply:** Thanks for your suggestion. We have revised the title of Figure S2, as indicated below:

[Figure]

**Figure S2** Romte sensing monthly Chl-*a* concentration (panel a) and total absorption coefficient at 443 nm (panel b) in April 2021. Data of Aqua-MODIS at resolution of 9 km were downloaded from https://oceancolor.gsfc.nasa.gov/